# The Cultivation of *Chelidonium majus* L. Increased the Total Alkaloid Content and Cytotoxic Activity Compared with Those of Wild-Grown Plants

**DOI:** 10.3390/plants10091971

**Published:** 2021-09-21

**Authors:** Valerija Krizhanovska, Inga Sile, Arta Kronberga, Ilva Nakurte, Ieva Mezaka, Maija Dambrova, Osvalds Pugovics, Solveiga Grinberga

**Affiliations:** 1Latvian Institute of Organic Synthesis, 21 Aizkraukles Str., LV-1006 Riga, Latvia; valerija@osi.lv (V.K.); inga.sile@farm.osi.lv (I.S.); maija.dambrova@farm.osi.lv (M.D.); osvalds@osi.lv (O.P.); 2Department of Dosage Form Technology, Riga Stradins University, 16 Dzirciema Str., LV-1007 Riga, Latvia; 3Field and Forest, SIA, 2 Izstades Str., LV-4126 Priekuli Parish, Cēsis County, Latvia; arta.kronberga@fieldandforest.lv; 4Institute for Environmental Solutions, “Lidlauks”, LV-4126 Priekuļi Parish, Cēsis County, Latvia; ilva.nakurte@vri.lv (I.N.); ieva.mezaka@vri.lv (I.M.); 5Department of Pharmaceutical Chemistry, Riga Stradins University, 16 Dzirciema Str., LV-1007 Riga, Latvia

**Keywords:** *Chelidonium majus*, alkaloids, flavonoids, LC/MS analysis, cytotoxic activity

## Abstract

The effect of cultivation practises on both the phytochemical profile and biological activity of aqueous ethanol extracts of *Chelidonium majus* L. was studied. Extracts were prepared from aerial parts of the same plant population collected in the wild and grown under organic farming conditions. Both qualitative and quantitative analyses of alkaloids and flavonoid derivatives were performed by LC/MS methods, and the cytotoxicity of lyophilised extracts was studied in B16-F10, HepG2, and CaCo-2 cells. Coptisine was the dominant alkaloid of extracts prepared from wild-grown plants, whereas after cultivation, chelidonine was the most abundant alkaloid. The total alkaloid content was significantly increased by cultivation. Ten flavonol glycoconjugates were identified in *C. majus* extracts, and quantitative analysis did not reveal significant differences between extracts prepared from wild-grown and cultivated specimens. Treatment with *C. majus* extracts resulted in a dose-dependent increase in cytotoxicity in all three cell lines. The extracts prepared from cultivated specimens showed higher cytotoxicity than the extracts prepared from wild-grown plants. The strongest cytotoxic effect of cultivated *C. majus* was observed in B16-F10 cells (IC_50_ = 174.98 ± 1.12 µg/mL). Cultivation-induced differences in the phytochemical composition of *C. majus* extracts resulted in significant increases in the cytotoxic activities of the preparations.

## 1. Introduction

Greater celandine, *Chelidonium majus* L. (*Papaveraceae* Juss), is a valuable medicinal plant that is widely distributed throughout Europe, Asia, Northwest Africa, and North America [1]. In Latvia, it is considered a native species occurring throughout the country from solitarily specimens to dense growths [2]. In traditional medicine, *C. majus* has been used to treat bile and liver disorders [3]. Fresh latex from plants has been used externally for the treatment of warts, corns, fungal infections, eczema, and tumours of the skin [4,5]. In Latvian folklore materials, fresh latex and tea made from *C. majus* were reported to be used for treating diarrhoea, eye problems, and skin diseases such as lichen and warts [6]. The treatment of ophthalmological problems and gastrointestinal and skin disorders are mentioned among many other ethnobotanical studies across Europe [7,8,9,10]. The European Medicines Agency (EMA) has proposed two possible therapeutic indications in the monograph on *Chelidonii herba*: for symptomatic relief of digestive disorders such as dyspepsia and flatulence (oral intake), as well as for treatment of warts, calluses, and corns (cutaneous use) [11]. However, these indications were not supported due to a lack of information on clinical safety. From a research point of view, this plant is still very interesting because it is widely used in folk medicine, but it has not yet acquired the status of an officially approved and evidence-based herbal medicine.

This species is known to produce a broad range of secondary metabolites, ensuring its therapeutic properties. The main constituents of *C. majus* responsible for biological properties are isoquinoline alkaloids such as chelidonine, chelerythrine, sanguinarine, coptisine, berberine, allocryptopine, and protopine [1]. They are reported to have anti-inflammatory, antimicrobial, antibacterial, antiviral, immunomodulatory, anticancer, choleretic, hepatoprotective, and analgesic properties [1,3]. *C. majus* alkaloids are a subject of interest due to their cytotoxic effects against various types of cancer cell lines [12,13,14]. The well-known product Ukrain^®^, a preparation consisting of a mixture of *C. majus* alkaloids, is marketed for its anticancer properties. However, many previous clinical studies are considered untrustworthy [15]. Most in vitro anticancer activity studies of *C. majus* refer to sanguinarine, chelidonine, chelerythrine, and berberine. Sanguinarine, which interacts strongly with DNA, has been shown to be the most potent anticancer agent obtained from *C. majus*. The IC50 values of sanguinarine in leukaemia cell lines are reported to be up to 0.10 µM [12] and 0.2 µM in human keratinocyte (HaCaT) cell lines [14]. Chelerythrine, berberine, and chelidonine are also active, but are less potent as cytostatic agents [12].

Comprehensive reports on the alkaloid profile of *C. majus* [16,17,18,19] are available. More than 50 alkaloids have been detected in greater celandine [17,20]. Quantitative analyses of the main alkaloids chelerythrine, sanguinarine, and coptisine in *C. majus* extracts were performed by HPLC-DAD and LC–MS/MS, and tentative identification of minor alkaloids was performed with data from the literature [16,19,21,22,23]. In contrast, data on flavonoid composition and content in *C. majus* are fragmented and mainly qualitative. Grosso et al. [16] quantified the flavonoid content with HPLC-DAD for the first time, and MRM methods were used for the determination of quercetin and phenolic acids [23].

Greater celandine, like other wild plants, shows interesting features with potential commercial viability. The market demand for biologically active ingredients from plants is increasing, and the cultivation of medicinal plants offers several benefits over collection of wild plants, e.g., reliable supply, standardised and improved production, and certainty of botanical identity. It is well known that the content of biologically active components of celandine is significantly affected by growing conditions [1,24]. Therefore, it is important to assess the influence of cultivation practices (growing in the wild or under organic farming conditions) on the phytochemical composition of *C. majus* populations.

The aim of this study was to investigate how growing conditions affect both the phytochemical compositions and cytotoxic activities of aqueous ethanol extracts of *C. majus*. Aerial parts of wild populations of *C. majus* originating from different regions of Latvia were harvested. Plantlets from the same wild populations were planted and cultivated under organic farming conditions. Aqueous ethanolic extracts were prepared from both wild-grown and cultivated plants. High-resolution mass spectrometry was applied for the identification of phytochemical compounds, and quantitative analyses of major components were performed by UPLC–MS/MS. CaCo-2, HepG2, and B16-F10 cell lines were selected for cytotoxicity analysis because in traditional medicine, aerial parts of *C. majus* have often been used to treat gastric and liver diseases, and locally for various skin disorders [1,17]. The cytotoxic activities of the lyophilised extracts were determined in these three cancer cell lines.

## 2. Results and Discussion

### 2.1. Alkaloid Profile and Quantitative Analysis of Aqueous Ethanol C. majus Extracts

LC/MS-TOF analyses of aqueous ethanol *C. majus* extracts revealed the presence of 12 alkaloids. The identities of chelidonine, sanguinarine, and chelerythrine were confirmed with available reference standards. Tentative identification of other alkaloids was performed by comparison of their chromatographic retention times and detected m/z values with literature data [19,25]. A summary of the identification results is shown in Table 1.

The results of quantitative determinations of major alkaloids revealed coptisine as the predominant compound in *C. majus* extracts prepared from wild-grown celandine (Table 2, Appendix A Appendix A).

Coptisine, as the predominant alkaloid, was also found by Sárközi et al. [21]; however, the extracts in their study were prepared using methylene chloride. Chelerythrine and sanguinarine were found to be the dominant alkaloids in methanol extracts [19]. In aqueous ethanol extracts, coptisine was found to be the dominant alkaloid; moreover, an increase in ethanol content from 25% to 45% resulted in a more than fivefold increase in coptisine recovery [26].

Our study shows that the chelidonine content in extracts of cultivated specimens was approximately four times higher than that in wild-grown plant preparations, and chelidonine became the dominant alkaloid. It should be noted that the variability of chelidonine content in extracts prepared from both cultivated and wild-grown *C. majus* specimens was very high (RSD > 50% in both cases, n = 5, Appendix A Appendix A). The total content of alkaloids in extracts prepared from cultivated *C. majus* specimens was significantly higher than that in extracts prepared from wild-grown *C.majus* specimens (Table 2). The contents of both sanguinarine and chelerythrine were significantly increased in extracts prepared from cultivated *C. majus* specimens. Coptisine, berberine, and allocryptopine contents also showed increasing tendencies in extracts of cultivated *C. majus* plants; however, the concentration variability between individual samples was still very high, and the differences were not statistically significant.

Wide variation in the alkaloid content in *C. majus* has been reported previously by other authors [18,27,28]. Many factors, such as genotype, plant age, developmental phases, harvesting time, and environmental conditions, can affect the alkaloid content in raw plant material.

### 2.2. Flavonoid Profile and Quantitative Analysis of Aqueous Ethanol C. majus Extracts

*C. majus* contains minor amounts of quercetin, kaempferol, and isorhamnetin glycoconjugates [16,29]. Recently, a small phenolic acids content [23] was discovered. Several studies [18,24] have focused on the total phenolics content or total flavonoids content. Mass spectrometry was sporadically applied for the identification of individual compounds [16].

To achieve nontarget identification of flavonoid derivatives, we screened the corresponding aglycone masses of quercetin, isorhamnetin, and kaempferol on an HRMS instrument and then performed MRM analyses on a tandem mass spectrometer (Appendix A Appendix A). The identified key flavonols quercetin, kaempferol, and isorhamnetin were found to be present in various glycosylated forms. In contrast to roots showing the presence of only quercetin aglycone [23], in aerial parts we identified 10 mono-, di-, and triglycosides of flavonols, as shown in Table 3.

Three intense peaks for flavonol 3-O-diglycosides consisting of both rhamnoside and glucoside fragments were observed. Peak **4** yielded a precursor ion with an m/z 611 [M + H]^+^ along with a fragment with m/z 465 for the loss of rhamnosyl, indicating that it is quercetin 3-O-rhamnosylglucoside, while peaks **7** and **9** with similar fragmentation patterns (595→449 and 625→479) were identified as kaempferol 3-O-rhamnosylglucoside and isorhamnetin 3-O-rhamnosylglucoside. The identities were confirmed by comparison with reference standards.

Along with monoglycosides and diglycosides (peaks **4****–10**), carbohydrate residues with three saccharide moieties were identified in extracts. Peak **2** showed a precursor ion at m/z 757 [M + H]^+^ (C_33_H_40_O_20_), and its MS/MS spectrum presented a product ion at m/z 611 attributed to the elimination of a glycosyl residue and a product ion at m/z 449 produced after loss of a rutinoside residue. Based on aglycone formation at m/z 287, this compound was tentatively identified as kaempferol 3-O-rhamnosylglycoside-7-O-glucoside [30]. Similarly, quercetin 3-O-triglycoside consisting of one rhamnosidylglycoside and one glucoside (peak **1**) and isorhamnetin 3-O-rhamnosylglycoside-7-O-glucoside (peak **3**) were identified. The identification of peak **3** was based on the MRM parent scan, as HR full scan mass spectra did not yield protonated molecular ions. These three glycosides (peaks **1**, **2**, and **3**) were formed from the same carbohydrates attached to the same positions of the three different flavonols and have not been previously reported in *C. majus*.

Two peaks with molecular masses equivalent to that of kaempferol diglycoside were detected (peaks **6** and **7**). Although both molecular and aglycone ions coincided, retention times were different. The later eluting peak (**7**) was identified as kaempferol-3-rutinose by comparison with the reference compound.

Quantitative analyses of flavonoids showed the predominance of rutinoside-type glycoconjugates (Table 4, Appendix A Appendix A), similar to the findings of Grosso et al. [16] and Parvu et al. [29]. Quercetin 3-O-rutinoside (rutin) was the predominant compound in extracts prepared from both wild-grown and cultivated *C. majus* specimens. The next most abundant flavonol glycosides were isorhamnetin 3-O-rutinoside and kaempferol 3-O-rutinoside. The total flavonol glycoside content was slightly higher in extracts prepared from cultivated *C. majus* specimens; however, the major contribution was the increase in rutin content, and the changes were not statistically significant.

### 2.3. Cytotoxic Activity of Extracts from C. majus

The cytotoxic activities of extracts towards B16-F10, HepG2, and CaCo-2 cell lines were evaluated by determination of IC_50_ values in the MTT assay. Treatment with *C. majus* extracts resulted in dose-dependent increases in cytotoxicity in all three cell lines (Figure 1). The strongest cytotoxic effect of *C. majus* was observed in B16-F10 cells. The IC_50_ values of the cultivated samples on B16-F10 cells ranged between 174.98 ± 1.12 µg/mL and 318.42 ± 1.08 µg/mL. HepG2 and CaCo-2 cells were less sensitive than the melanoma cells, and the IC50 values of the cultivated samples ranged between 226.46 ± 1.66 µg/mL and 448.75 ± 1.34 µg/mL and from 291.07 ± 1.10 µg/mL to 406.44 ± 1.08 µg/mL, respectively (Table 5).

The cytotoxicity data of this study showed activities for aqueous ethanol extracts that are lower than those previously reported. Thus, the IC_50_ value for the methanol extract on CaCo-2 cells was 166.06 ± 15.71 µg/mL, and that on HepG-2 cells was 144.81 ± 15.03 µg/mL [31]. In the study by Fadhil, [32], the cytotoxic effect (IC_50_ = 282.86 µg/mL) of *C. majus* on HepG-2 cells was in line with the present results. *C. majus* cultivated under controlled environmental conditions exhibited higher cytotoxic activity against all studied cell lines.

The results of the present study showed that the cytotoxic effect in different cell lines varies depending on the alkaloid assessed. In experiments with HepG2 cells treated with various *C. majus* extracts, the strongest correlation was between the IC_50_ values and the contents of chelerythrine, sanguinarine, and chelidonine. In the case of B16-F10 cells, there was a correlation between the IC_50_ values and sanguinarine, chelidonine, and total alkaloid contents. Furthermore, the IC_50_ values for CaCo-2 were correlated with the sanguinarine, chelerythrine, allocryptopine, and total alkaloid contents (Appendix A Appendix A). Although the amount of chelidonine differed by up to four times between cultured and wild-grown *C. majus* samples, this was not clearly reflected in the biological activity results. Our data provide additional evidence that some alkaloids, such as chelerythrine, sanguinarine, and berberine, are more important in causing apoptosis or stopping the proliferation of cancer cells. Unlike the previously mentioned alkaloids, chelidonine is a weak DNA intercalating agent and does not cause lethal mutations or DNA damage [1,33]. This relationship is very well demonstrated in the study [33], where all these substances were tested on different cancer cells and chelerythrine, sanguinarine, and berberine showed significantly higher cytotoxicity. In human pharyngeal squamous carcinoma cells (FaDu), the difference between the IC_50_ values of sanguinarine and chelidonine was more than 500 times [33]. In our study, the cytotoxic activities of wild-grown and cultivated *C. majus* extracts were evaluated for the first time in the B16-F10, HepG2, and CaCo-2 cell lines. Our results confirmed the cytotoxicity of *C. majus* extracts towards the studied cell lines, which indicates the usefulness of compounds found in the extracts for the treatment of different cancer types.

## 3. Materials and Methods

### 3.1. Chemicals and Reagents

HPLC gradient grade acetonitrile and formic acid were purchased from Sigma-Aldrich (Schnelldorf, Germany). The reference substances sanguinarine, chelerythrine, and chelidonine were purchased from Biosynth Carbosynth (Compton, UK), Alfa Aesar Chemicals (Heysham, UK), and Cayman Chemical (Ann Arbor, MI, USA), respectively. All flavonoid reference substances were purchased from PhytoLab (Vestenbergsgreuth, Germany).

### 3.2. Plant Materials and Preparation of Aqueous Ethanol Extracts

Aerial parts from five populations of *Chelidonium majus* were collected from the wild at the flowering stage (hereafter referred to as “wild”) for chemical analysis and biological activity testing during May 2019 (Appendix A Appendix A). Voucher specimens were deposited at the Institute for Environmental Solutions (IES) in Latvia under codes CHM01, CHM02, CHM03, CHM04, and CHM05. Ten randomly selected plantlets were also collected from the same five populations in 2019 and planted in an organically certified experimental field of IES (57°19’11.7” N 25°19’18.8” E, 115 m altitude). The plot size was 0.8 m^2^, and the plant spacing was 0.2 × 0.5 m. A year later, aerial parts were collected during the flowering stage from the same populations in the experimental field (hereafter referred to as “cultivated”). The collected plant material was dried at 55 °C for 14–29 h, and then the plant material was powdered. Powdered dried samples of *C. majus* were macerated with 70% ethanol solution in water at 1:10 w/v. Prepared solutions were incubated for 7 days in a dark, cool place and frequently shaken until extraction of the plant material was completed. Afterwards, the material was pressed, and the remaining solid was squeezed to remove all remaining solvent. The obtained solutions were clarified by decantation and centrifugation.

### 3.3. Preparation of Lyophilised Extracts

For the biological activity assays, the aqueous ethanol extracts were concentrated with a rotary evaporator and further lyophilised. The obtained powder was labelled and stored in a refrigerator at −20 °C prior to further analysis.

In vitro experiments with *C. majus* extracts were carried out using lyophilised plant material dissolved in dimethyl sulfoxide (DMSO). The final concentration of DMSO in each well did not exceed 0.5% (v/v).

### 3.4. HRMS Analysis

The plant extracts were analysed on a Shimadzu LCMS hybrid IT-TOF system combined with a Nexera X2 UPLC system. An Acquity UPLC BEH C18 (2.1 × 150 mm, 1.7 μm particle size) column was used with a flow rate of 0.4 mL/min. The column oven was set at 40 °C, and the sample injection volume was 1 μL. The mobile phase consisted of a combination of A (0.1% formic acid in water) and B (acetonitrile). Gradient: 2% B, 1 min—2% B, 4 min—5% B, 14 min—15% B, 36 min—50% B, 48 min—98% B, 55 min—98% B, 58 min—2% B, 60 min—2% B.

The adjusted operating parameters of the mass spectrometer were set as follows: detector voltage—1.5 kV, nebulizing gas (N2) flow—1.5 mL/min, mass scan range (m/z) —120 to 1000, and ion accumulation time—10 ms. LCMSsolution software was used to process LCMS data. UV/Vis spectra were recorded over the range 190 nm to 650 nm.

Aqueous ethanol extracts of dried plant material were injected into the chromatographic system without further processing.

### 3.5. UPLC–MS/MS Analysis of Alkaloids

UPLC separation of target compounds in plant extracts was performed on an Acquity BEH C18 column (2.1 × 50 mm, 1.7 μm, Waters) using a Waters Acquity UPLC system. A linear gradient from 5% acetonitrile in 0.1% aqueous formic acid to 98% acetonitrile was applied over 5 min. A Xevo TQ-Smicro tandem mass spectrometer (Waters) in positive electrospray mode was used for quantification. Multiple reaction monitoring (MRM) parameters are detailed in Appendix A Appendix A.

Aqueous ethanol extracts were diluted 10 or 100 times with 70% ethanol before MRM analysis. Calibration concentrations ranged from 1 ng/mL to 250 ng/mL for all analytes.

### 3.6. UPLC–MS/MS Analysis of Flavonoids

UPLC separation of flavonoid glycoconjugates in plant extracts was performed on an Acquity BEH C18 column (2.1 × 100 mm, 1.7 μm, Waters) using a Waters Acquity UPLC system. A linear gradient from 5% acetonitrile in 0.1% aqueous formic acid to 98% acetonitrile was applied over 12 min. A Xevo TQ-Smicro tandem mass spectrometer (Waters) in positive electrospray mode was used for quantification. Multiple reaction monitoring (MRM) parameters are detailed in Appendix A Appendix A. Aqueous ethanol extracts were diluted 100 times with reserpine (internal standard) solution (10 ng/mL) in 70% ethanol before MRM analyses. Calibration concentrations ranged from 50 ng/mL to 10 µg/mL for all analytes.

### 3.7. Cell Culture

The B16-F10 murine melanoma cell line (CRL-6475™), HepG2 human hepatocellular carcinoma cell line (HB-8065™), and CaCo-2 human colorectal adenocarcinoma cell line (HTB-37™) were purchased from ATCC^®^ (American Type Culture Collection, Manassas, VA, USA) and cultured in DMEM with Glutamax (Gibco, Darmstadt, Germany) supplemented with 10%–20% foetal bovine serum (FBS, Merck KGaA, Darmstadt, Germany) and 1% antibiotics (100 U/mL penicillin and 100 μg/mL streptomycin) at 37 °C in a humidified incubator under 5% CO_2_. After reaching 80% confluence, the cells were subcultured in 96-well plates at a final concentration of 10 × 10^4^ cells/mL (100 µL medium in each well).

### 3.8. Cytotoxicity Assay

To estimate the cytotoxicity of extracts against three cell lines, MTT (3-[4,5-diethylthiazol- 2-yl]-2,5-diphenyltetrazolium bromide) and a slightly modified method by Mosmann [34] were used. Cells were seeded in 96-well plates at a final concentration of 10 × 10^4^ cells/mL (100 µL medium in each well) and incubated overnight for adherence. Then, 100 µL of medium or extract dilution in medium (100 μg/mL–1200 μg/mL) was added to each well. The medium with the tested extracts was added at different concentrations. After 24 h incubation with the extracts, the medium was exchanged with 100 µL of MTT solution (1 mg/mL in PBS) and incubated for 2 h at 37 °C. Thereafter, the solution was aspirated, and isopropanol was added to each well to dissolve the formazan crystals formed during the incubation period. The plate was placed in a shaker for dissolution. The absorbance was determined spectrophotometrically at 570 nm using a reference wavelength of 650 nm on a Hidex Sense microplate reader (Hidex, Turku, Finland).

### 3.9. Statistical Analysis

The data obtained from the biological activity assay were analysed using the log (inhibitor) vs. response—variable slope (four parameters) analysis function and performed with GraphPad Prism (GraphPad, Inc., La Jolla, CA, USA) computer software. IC_50_ values were obtained from three independent experiments (n = 6) and are presented as the means ± SD. The quantitative results (content of flavonoids and alkaloids) are presented as the mean ± standard deviation (SD). Statistical analysis was performed using Student’s *t*-test (two tailed distribution, two sample equal variances). *p* < 0.05 was considered statistically significant.

## 4. Conclusions

The total content of alkaloids in aqueous ethanol extracts prepared from cultivated *C. majus* specimens was higher than that observed in extracts prepared from wild-grown plant material. Chelidonine, sanguinarine, and chelerythrine were the main contributors to the total increase in alkaloid content. The cultivation of *C. majus* did not significantly affect the total content of flavonol glycosides. The observed differences in the phytochemical compositions of the *C. majus* extracts resulted in significant increases in the cytotoxic activities of the preparations.

## Figures and Tables

**Figure 1 plants-10-01971-f001:**
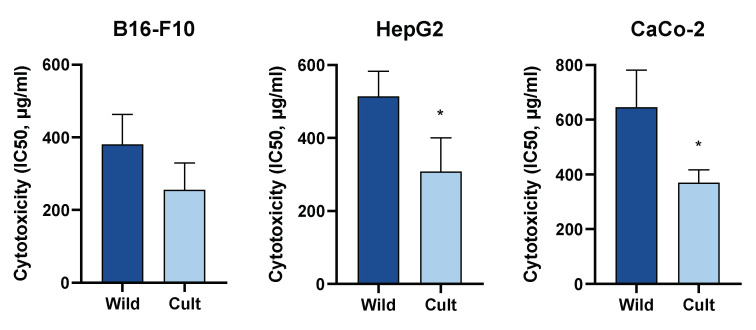
Differences in cytotoxic activities between cultivated and wild-grown *C. majus* on B16-F10, HepG2, and CaCo-2 cell lines measured by MTT assay. Values are the mean ± SD (n = 5). Differences between the measurements were tested using the Mann-Whitney U-test. * Significantly different from wild-grown *C. majus* (*p* < 0.05).

**Table 1 plants-10-01971-t001:** List of tentatively identified alkaloids in the ethanol extracts of aerial parts of *C. majus.*

RT, min	m/z	Compound	MW (Monoisotopic)	Calculated Elemental Composition
18.02	354.134	Protopine ^1^	353.126	C_20_H_19_NO_5_
18.81	354.133	**Chelidonine**	353.126	C_20_H_19_NO_5_
19.00	370.163	Allocryptopine ^1^	369.158	C_21_H_23_NO_5_
19.09	320.092	Coptisine ^1^	320.092	C_19_H_14_NO_4_^+^
20.2	370.165	Allocryptopine ^1^	369.158	C_21_H_23_NO_5_
20.33	340.118	Norchelidonine ^1^	339.111	C_19_H_17_NO_5_
20.56	340.153	Canadine ^1^	339.147	C_20_H_21_NO_4_
21.05	340.118	Norchelidonine ^1^	339.111	C_19_H_17_NO_5_
21.28	332.091	**Sanguinarine**	332.092	C_20_H_14_NO_4_^+^
21.87	336.122	Berberine ^1^	336.124	C_20_H_18_NO_4_^+^
23.46 ^2^	348.122	**Chelerythrine**	348.124	C_21_H_18_NO_4_^+^
23.46 ^2^	382.128	6,10-Dihydroxyl chelerythrine ^3^	382.129	C_21_H_20_NO_6_^+^

^1^ [19], ^2^ overlapping chromatographic signals, ^3^ [25], **bold**—identified by comparison with reference compounds.

**Table 2 plants-10-01971-t002:** Content of alkaloids (µg/g of dry material) in the ethanol extracts of aerial parts of wild-grown and cultivated *C. majus.*

Compound	Average Alkaloid Content (n = 5)		*p* Value
	Wild 2019	Cultivated 2020	
Sanguinarine	1.9 ± 2.1	12.8 ± 3.6	0.0004
Chelerythrine	3.5 ± 1.3	17.5 ± 8.5	0.007
Chelidonine	63.6 ± 35.4	252.2 ± 133.2	0.02
Coptisine ^1^	138.5 ± 35.6	143.5 ± 32.2	0.8
Berberine ^1^	9.4 ± 6.6	12.8 ± 8.4	0.6
Allocryptopine ^1^	5.2 ± 3.0	11.9 ± 7.4	0.1
Total Content	222.0	450.6	0.02

^1^ Coptisine, berberine, and allocryptopine quantified as chelidonine.

**Table 3 plants-10-01971-t003:** List of tentatively identified flavonoids in the ethanol extracts of aerial parts of *C. majus.*

Peak #	RT, min	Characteristic ions ^1^ ESI^+^, m/z	Characteristic ions ^1^ ESI^−^, m/z	Compound	MW (Monoisotopic)	Calculated Elemental Composition ^2^	Parent Scan of Aglycone Fragment Ion, m/z
**1**	11.2	627.144		Quercetin Triglycoside	772.206	C_33_H_40_O_21_	773, 627,465 (303) ^3^
**2**	13.0		755.201	Kaempferol Triglycoside	756.211	C_33_H_40_O_20_	757, 611, 449 (287)
**3**	13.6	479.119		Isorhamnetin Triglycoside	786.222	C_34_H_42_O_21_	787, 641, 479 (317)
**4**	16.8	611.159	609.143	**Quercetin 3-O-Rutinoside**	610.153	C_27_H_30_O_16_	611, 465 (303)
**5**	17.0	465.101		**Quercetin 3-O-Galactoside**	464.100	C_21_H_20_O_12_	465 (303)
**6**	17.4	287.056	593.147	Kaempferol Diglycoside	594.158	C_27_H_30_O_15_	595, 449 (303)
**7**	18.4	287.054	593.147	**Kaempferol 3-O-Rutinoside**	594.101	C_27_H_30_O_15_	595, 449 (287)
**8**	18.8	449.104	447.089	Kaempferol Glucoside	448.101	C_21_H_20_O_11_	449 (287)
**9**	19.0	625.173	623.158	**Isorhamnetin 3-O-Rutinoside**	624.169	C_28_H_32_O_16_	626, 279 (317)
**10**	19.4	479.119	477.102	Isorhamnetin Glycoside	478.111	C_22_H_22_O_12_	479 (317)

^1^ HRMS data, ^2^ mass difference within ± 5 mDa, ^3^ m/z of aglycone fragment in brackets, **bold**—identified by comparison with reference compounds.

**Table 4 plants-10-01971-t004:** Content of flavonoids (µg/g of dry material) in the ethanol extracts of aerial parts of wild-grown and cultivated *C. majus.*

Compound	Average Flavonoid Content (n = 5)		*p* Value
	Wild 2019	Cultivated 2020	
Kaempferol	13.1 ± 9.2	6.9 ± 4.2	0.2
Isorhamnetin	8.8 ± 6.7	4.0 ± 1.6	0.2
Quercitrin	1.4 ± 1.2	2.0 ± 0.9	0.4
Isorhamnetin 3-O-Rutinoside	1612.7 ± 722.9	1857.2 ± 326.0	0.5
Kaempferol 3-O-Rutinoside	653.8 ± 377.4	600.7 ± 216.4	0.8
Quercetin 3-O-Rutinoside	3007.2 ± 1270.1	4385.1 ± 1150.8	0.1
Quercetin 3-O-Galactoside	220.2 ± 269.9	195.9 ± 114.0	0.9
Kaempferol Glucoside ^1^	135.9 ± 130.8	53.5 ± 12.9	0.2
Total	5653.1	7105.3	0.3

^1.^ Kaempferol glucoside—quantified as luteolin 7-O-glucoside.

**Table 5 plants-10-01971-t005:** Inhibitory effects on the growth of human hepatocellular carcinoma (HepG2), murine melanoma (B16-F10), and human colorectal adenocarcinoma (CaCo-2) cells.

Sample	IC_50_ (µg/mL) ± SD		
	HepG2	B16-F10	CaCo-2
**Wild 2019**			
CHM01	422.67 ± 1.09	264.85 ± 1.13	>500
CHM02	>500	354.81 ± 1.22	>500
CHM03	>500	496.59 ± 1.05	>500
CHM04	>500	389.94 ± 1.12	>500
CHM05	461.32 ± 1.13	394.46 ± 1.08	>500
**Cultivated 2020**			
CHM01	351.56 ± 1.38	279.25 ± 1.08	361.41 ± 1.84
CHM02	241.55 ± 1.22	174.98 ± 1.12	291.07 ± 1.10
CHM03	272.27 ± 1.16	325.84 ± 1.20	406.44 ± 1.08
CHM04	226.46 ± 1.66	318.42 ± 1.08	389.94 ± 1.20
CHM05	448.75 ± 1.34	180.30 ± 1.54	400.87 ± 1.18

## Data Availability

Data available on request.

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
