# Peer review of "The Cultivation of Chelidonium majus L. Increased the Total Alkaloid Content and Cytotoxic Activity Compared with Those of Wild-Grown Plants"

_plants, 2021, doi:10.3390/plants10091971_

Round 1

Reviewer 1 Report

Grinberga and coworkers have investigated the effect of cultivation method of C. majus on the phytochemical profile and biological activity their aqueous ethanol extracts. HRMS and LC-MS were used to analyze the qualitative and quantitative of alkaloids respectively. HepG2, and CaCo-2 cells had been used to evaluate the cytotoxic activities of extracts. Their results showed that the total content of alkaloids in aqueous ethanol extracts prepared from cultivated C. majus was higher than that from wild-grown. It’s an interesting result. Because people always think the wild herbs are much better for drug uses. This reviewer recommend its publication after minor revisions.

  • Title, “exerts” –“has”
  • Page 4, table 2, The total content of cultivated 2020 is 450.

Reviewer 2 Report

In this MS, the authors studied the total alkaloid content and cytotoxic activity of the the cultivated and wild greater celandine Chelidonium majus L.. And they indicated that he cultivated greater celandine C. majus exerts a higher total alkaloid content and cytotoxic activity than wild plants. The topic is very interesting and novel. While there were many shortcomings and errors, which were following as:

  Q1: In this study, two types of C. majus, i.,e., the cultivated and wild greater celandine C. majus were studied. While the cultivated greater celandine was got through planting the wild C. majus in pots just for one year, not cultivated for many years! So it is not suitable or rigorous to say the cultivated greater celandine C. majus.

  Q2: Table 1 - Table 4: The titles were unclear. What plant tissues were used in this study?

  Q3: Table 2 and Table 4: It is not suitable to give P2 values, it should be given that the t and P values of the Student's-t test.

  Q4: Table 3: Confused data results to show MS/MS fragments (ESI+)! Just give order-list data! E.g., 773>627>465>303. And in this table, the data wasn’t show cultivated or wild greater celandine.

  Q5: 2.3. Cytotoxic activity of extracts from C. majus: “The IC50 values of the cultivated samples on B16-F10 cells ranged between 174.98 ± 1.12 µg/ml and 318.42 ± 1.08 µg/ml. HepG2 and CaCo-2 cells were less sensitive than melanoma cells, and the IC50 values of the cultivated samples ranged between 226.46 ± 1.66 µg/ml and 448.75 ± 1891.34 µg/ml and from 291.07 ± 1.10 µg/ml to 406.44 ± 1.08 µg/ml, respectively”  Confused the "range" from one range to another range!

  Q6: 3.2. Plant materials and preparation of aqueous ethanol extracts: “Ten randomly selected plantlets were also collected from the same five populations in 2019 and planted in an organically certified experimental field of IES (57°19'11.7"N 25°19'18.8"E, 115 m altitude). The plot size was 0.8 m2, and the plant spacing was 50 x 20 cm. A year later, aerial parts were collected during the flowering stage from the same populations in the experimental field”  After these treatments of the five populations of C. majus collected from the wild, they became the cultivated C. majus specimens? Not suitable or rigorous to say cultivated greater celandine C. majus.

  Q7: 3.8. Cytotoxicity assay: The MTT assay was not introduced in details.

Reviewer 3 Report

In this article it were described the alkaloid content and cytotoxic activity effects of Chelidonium majus L. However there are several aspects that should be ameliorated:

  • The title it was confusing due to the term "greater".
  • In the abstract some values of IC50 with the different cell types could be added in order to be more understandable the magnitude of the findings.
  • In the section 1. introduction (lines 89-90) the authors indicated that three cancer cell lines but forget to indicate which were these cell lines). Which was the reason to select the three cell lines?
  • In the section 2. results and discussion the authors referred that evaluated the composition, did the authors performed tested some of the major individual compounds in the cancer cell lines?
  • In tables  2 and 4 it seems that the values could be present with one decimal point.
  • The figure 1 it was a little difficult to understand as some of the IC50 values are overlapped. Eventually, a table (instead these figure) or a different type of graph could be more appropriate.
  • In the results and dicussion section, the results should be whenever possible to be compared with the literature, e.g. the authors could compare with this study in which was also used HepG2 cells: Fadhil, Y. B., Alsammarraie, K. W., Mohaimen, N. A., & Mohammed, Z. Y. (2018). In Vitro Cytotoxic Activity of Chelidonium majus extract using Different Types of Cell Lines. Int. J. Curr. Microbiol. App. Sci7(1), 1767-1775.
  • In section 3. Materials and Methods, more specifically in the 3.9. Statistically analysis should be described the tests used in the comparison between the two groups.
  • In the conclusion section, the authors emphasize that the values of rutin show a tendency to be higher but I can not see these values. In addition, for the isorhamnetin 3-O-rutinoside as no statistically significant differences were found (P values of 0.5) I think that these should not be emphasized. As a suggestion please give more importance to the cytotoxicity activity found.

Round 2

Reviewer 2 Report

This version of this manuscript has been revised based on the reviewers' comments.